# Effect of Physicochemical Properties of Carboxymethyl Cellulose on Diffusion of Glucose

**DOI:** 10.3390/nu13051398

**Published:** 2021-04-21

**Authors:** Elisabeth Miehle, Stephanie Bader-Mittermaier, Ute Schweiggert-Weisz, Hans Hauner, Peter Eisner

**Affiliations:** 1TUM School of Life Sciences, Technical University of Munich (TUM), 85354 Freising, Germany; 2Fraunhofer Institute for Process Engineering and Packaging (IVV), 85354 Freising, Germany; stephanie.mittermaier@ivv.fraunhofer.de (S.B.-M.); ute.weisz@ivv.fraunhofer.de (U.S.-W.); peter.eisner@ivv.fraunhofer.de (P.E.); 3Institute of Nutritional and Food Sciences, University of Bonn, 53113 Bonn, Germany; 4Else Kröner-Fresenius-Center for Nutritional Medicine, School of Life Sciences, Technical University of Munich, 85354 Freising, Germany; hans.hauner@tum.de; 5Institute of Nutritional Medicine, School of Medicine, Technical University of Munich, 80992 Munich, Germany; 6ZIEL-Institute for Food & Health, TUM School of Life Sciences Weihenstephan, Technical University of Munich, 85354 Freising, Germany; 7Faculty of Technology and Engineering, Steinbeis-Hochschule, 01069 Dresden, Germany

**Keywords:** in vitro glucose release, sodium carboxymethyl cellulose, soluble dietary fiber, critical concentration, viscosity, diffusion coefficient, molecular weight

## Abstract

Soluble dietary fibers (SDF) are known to reduce the post-prandial plasma glucose levels. However, the detailed mechanisms of this reduced glucose release in the human gut still remain unclear. The aim of our study was to systematically investigate the effect of different types of SDF on glucose release in an in vitro model as a prerequisite for the selection of fibers suitable for application in humans. Three types of carboxymethyl cellulose (CMC) were used to investigate the correlations between fiber concentration, molecular weight (*M_W_*), and viscosity on diffusion of glucose using a side-by-side system. CMC solutions below the coil overlap (*c**) influenced the glucose diffusivity only marginally, whereas at concentrations above *c** the diffusion of glucose was significantly decreased. Solutions of lower *M_W_* exhibited a lower viscosity with lower glucose diffusion compared to solutions with higher *M_W_* CMC, attributed to the higher density of the solutions. All CMC solutions showed a systematic positive deviation from Stokes-Einstein behavior indicating a greater rise in viscosity than reduction in diffusion. Therefore, our results pave the way for a new approach for assessing glucose diffusion in solutions comprising dietary fibers and may contribute to further elucidating the mechanisms of post-prandial plasma glucose level reduction.

## 1. Introduction

One of many health benefits of soluble dietary fibers (SDF) includes the reduction of post-prandial blood glucose level [1,2,3], which is of significance for people with impaired glucose tolerance or overt Type 2 diabetes mellitus. Several studies indicated that some types of SDF—like carboxymethyl cellulose (CMC)—reduce sugar absorption by altering the viscosity of the gut content [1,4,5,6,7,8].

The transfer of low molecular weight (*M_W_*) substances like glucose within the gut can be attributed to two different mechanisms, i.e., namely diffusion and convection [9]. It is well known that viscosity influences both diffusion and convection. However, the extent of such effects remains unclear. Higher viscosity of the chyme is considered to lead to decreased diffusion of low molecular substances from the gut lumen to the enterocytes and to widen the unstirred mucosal layer [10,11]. Besides, convection driven by intestinal contractions is also reduced by enhanced viscosity leading to a lower absorption of low *M_W_* substances [9,12,13] and resulting in reduced post-prandial plasma glucose levels in the case of glucose absorption [1]. Viscosity of the chyme is affected by the viscosity of the consumed food product, but also by dilution and/or electrolytes mediated by gastrointestinal secretion [3,14,15]. Viscosity of food products can be easily modified by incorporating dietary fiber ingredients in different concentrations and *M_W_* [16].

Thus, when considering dietary fiber solutions, the viscosity of a solution depends on the one hand on the volume occupied by each of the polymer coils, also known as the intrinsic viscosity, which is mainly influenced by the conformation of dietary fiber molecules, the *M_W_* and the type of solvent used for solubilization. On the other hand, it depends on the number of coils present in the solution, equating to the concentration [17].

Solutions of soluble dietary fibers with concentrations below the critical concentration (*c**) show Newtonian and above shear thinning flow behavior. According to Morris, Cutler, Ross-Murphy, Rees and Priceet al. [17] the flow behavior can be explained by the entanglement model for “random coil” polysaccharide solutions, whereby the formation of an entangled network characterized by coil overlap leads to a strong increase in viscosity. The onset of coil overlap can be induced by the critical concentration *c** at a given *M_W_* of a polysaccharide [17]. As solutions with concentrations above *c** show shear thinning behavior, viscosity has to be extrapolated to zero shear rate for a better comparison.

The zero-shear viscosity (*η*) and the diffusion coefficient (*D*) of the continuous phase in a solution are indirectly related according to the Stokes-Einstein (SE) relation:(1)$$D=\frac{k_{B}T}{6\pi R\eta }$$
where by *k*_B_ is the Boltzmann constant, *T* is the absolute temperature and *R* is the hydrodynamic radius of the sphere.

Deviations from SE relation regarding the diffusivity in viscous polymer solutions have been observed in the past [18,19,20,21]. The entanglements and differences in viscosity occurring in the polysaccharide solutions could be a possible reason for the deviations [20,21].

Despite the findings that the presence of SDF in the gut content increases viscosity and leads to altered glucose absorption, it is not well understood to what extent individual parameters of SDF like *M_W_*, concentration and viscosity have an effect on glucose release [22,23].

Therefore, the main objective of this work was to investigate the impact of *M_W_*, concentration, particularly *c**, and viscosity of CMC, which distinguish solely in their *M_W_*, on glucose diffusion using an in vitro side-by-side system. To examine the impact of *c** on the diffusivity of glucose, we conducted experiments with solutions in concentrations below and above *c**. We used three types of food grade sodium CMC differing in their *M_W_*, in consistent quality and well-characterized due to the non-natural occurrence in comparison to natural fibers like beta-glucan, which are subject to natural fluctuations for example in branching or molecular weight. CMC is used in a wide range of food products in various concentrations to achieve particular effects as water binding or thickening according to their physicochemical properties [24].

Medium and highly substituted CMC—induced by a high content of carboxymethyl groups—have an excellent solubility over a wide temperature range (0 °C–100 °C). They also form clear and smooth solutions, with a high electrolyte, temperature and pH stability [24] and exhibit shear thinning flow behavior at higher concentrations [25,26,27]. 

The flow behavior of the three different types of CMC in aqueous solutions were investigated by steady-state rheological measurements, and the *c** were determined using the zero-shear viscosities according to the model of Cross [28]. Both diffusion coefficients of glucose in the CMC solutions under static conditions as well as convection release behavior under stirred conditions were determined.

## 2. Materials and Methods

### 2.1. Materials 

Three types of food grade sodium carboxymethyl celluloses (CMC) were purchased from Ashland Industries Europe GmbH (Schaffhausen, Switzerland) and were denoted as CMC-L (Blanose™ 9LCF, weight-average molecular weight (*M_W_*) was 100 kDa), CMC-M (Blanose™ 9M31 F, *M_W_* of 395 kDa) and CMC-H (Blanose™ 9H4F, *M_W_* of 725 kDa). The degree of methylation (i.e., the amount of substituted OH-groups) of each sample ranged from 0.80 to 0.95, which corresponds to an average of 30% substituted OH-groups. All other reagents and chemicals used were of analytical grade and were supplied by VWR (Darmstadt, Germany) or Chemsolute (Th. Geyer GmbH & Co. KG, Renningen, Germany). 

### 2.2. CMC Solution Preparation

#### 2.2.1. Solutions for Determining the Critical Concentration c*

CMC solutions with concentrations ranging from 0.03 to 7 g/100 g were prepared as described by Kpodo et al. [29] with some modifications. Sodium phosphate buffer (0.1 mol/L, pH = 7.2) with 0.1 g/100 g sodium azide to prevent microbial growth was mixed with the appropriate amount of CMC. For complete solubilization, the solutions were gently stirred using a magnetic stirrer at room temperature overnight for 16 h. Prior to the rheological determination of the zero-shear viscosity, the solutions were equilibrated to 37 °C in a water bath for 20 min and degassed by ultrasonication at 37 °C for 3 min. Each solution was prepared at least in duplicate without any further storage between production and equilibration.

#### 2.2.2. Solutions for Determining the Glucose Diffusion

The determined critical concentration (*c**) of each type of CMC was used to calculate the concentrations for determining in vitro glucose release (Section 2.3), using the listed factors shown in Table 1 (three concentrations below and three concentrations above *c**). The obtained CMC concentrations were 0.28 to 4.52 g/100 g for CMC-L, 0.09 to 1.50 g/100 g for CMC-M and 0.05 to 0.80 g/100 g for CMC-H.

The respective amount of the different types of CMC was hydrated using distilled water in half of the final weight and stirred at 180 rpm and room temperature for two hours. Electrolytes were added in the final electrolyte concentration of the digestion mixture according to Minekus et al. [30] along with 0.1 mol/L glucose and 0.1 g/100 g NaN_3_ and distilled water added to reach the final concentration. Stirring was continued under the same conditions for 14 h to ensure complete dissolution of the CMC. The concentrations of electrolytes in the final CMC solutions were 7.27 mmol/L KCl, 1.12 mmol/L KH_2_PO_4_, 51.3 mmol/L NaHCO_3_, 32.6 mmol/L NaCl, 0.21 mmol/L MgCl_2_ * 6 H_2_O, 0.15 mmol/L (NH_4_)_2_CO_3_, 0.53 mmol/L CaCl_2_. For performing the in vitro glucose release measurements, the pH of the samples were adjusted to 7.0 ± 0.1 with 6 mol/L HCl or 3 mol/L NaOH. A sample without CMC comprising distilled water, electrolytes in the same concentration as described before, 0.1 mol/L glucose and 0.1 g/100 g NaN_3_ was used as blank.

### 2.3. In Vitro Glucose Release (IVGR) Measurement and Glucose Determination

#### 2.3.1. IVGR

The IVGR was determined for all concentrations and types of CMC displayed in Table 1 without stirring to determine diffusion coefficients. The highest concentration (c_6_) of each CMC was further used to evaluate the glucose release under convection. For the IVGR, a side-by-side diffusion system (SES GmbH, Bechenheim, Germany) (Figure 1) was used. A dialysis membrane (regenerated cellulose) with a molecular weight cut-off of 12–14 kDa (SERVA Electrophoresis GmbH, Heidelberg, Germany) and an effective area of mass transfer (*A*) of 1 cm^2^ (diameter = 1.128 cm) was applied.

The side-by-side system was heated to 37.0 ± 0.1 °C by a thermostatic circulating water bath and kept constant during IVGR. The membrane was placed between two Teflon foam gaskets for fixation between the donor and the receptor cell. 4 mL of phosphate buffer (174.6 mmol/L, with 0.1 g/100 g NaN_3_, adjusted to pH 7) was used as receptor fluid to receive the same molarity as the blank sample. After addition of the receptor fluid, the two sampling ports were closed and equilibration at 37.0 °C ± 0.1 °C was carried out for at least 10 min under constant stirring (300 rpm), which was pursued during IVGR.

An aliquot of 15 mL of the sample (donor fluid, pH 7) was equilibrated at 37 °C for 10 min in a water bath. 4 mL of the heated test sample was transferred to the donor cell. Aliquots of 100 µL of the receptor fluid were taken at 1, 5, 10, 20, 30, 40, 60, 90, 120, 180, 240, 360, 480 and 1440 min, respectively, and immediately replaced by 100 µL of phosphate buffer. After 24 h, also 100 μL of the donor fluid was sampled. The amount of glucose, which was reduced by replacing the dissolved glucose solution by phosphate buffer after sampling, was taken into account by the following calculation for the next sampling period: (2)$$c_{n^{*}}{=c}_{n}+\overset{n-1}{\underset{i=1}{\sum }}c_{i}\times 0.025$$
where *c*_n_ is the glucose concentration of the sample at a certain sampling time, in total 14 samples (*c*_1_–*c*_14_), and *c*_n*_ is the calculated concentration of the sample *c*_n_.

The IVGR of each CMC concentration and type of CMC was conducted at least in duplicate with freshly prepared solutions using two different side-by-side diffusion systems. For convection experiments, the donor fluid was stirred at 150 rpm throughout IVGR.

#### 2.3.2. Glucose Determination

The glucose concentration of the samples was determined using an enzymatic test kit for D-glucose (D-Glucose, Food & Feed Analysis, R-Biopharm AG, Darmstadt, Germany) following the manufacturer’s instructions with slight modifications for application using a 96-well plate. A volume of 200 µL of the diluted sample was mixed with 100 µL of the assay reagent and 2 µL of mixed enzyme solution. The reaction mixture was incubated at room temperature under shaking conditions until the reaction has stopped (no change in absorption in a time range of 15 min). Absorbance was measured at a wavelength of 340 nm using a microplate reader (BioTek Instruments GmbH, Bad Friedrichshall, Germany). The glucose concentration of the samples was calculated based on a standard curve with four measurement points (10, 30, 50, 100 mg/L) in each measurement. From each sample, the glucose concentration was determined in duplicate.

### 2.4. Glucose Release Kinetics and Determination of the Diffusion Coefficient

#### 2.4.1. Glucose Release Kinetics

The glucose release kinetics were fitted using a non-linear first-order kinetic described in Equation (3) according to Macheras et al. [31] and Naumann et al. [32]:(3)$$c_{t}{=c}_{f}\;[1-\exp\left(-kt\right)]$$
where *c*_f_ is the concentration of glucose after reaching equilibrium, *t* is the time in minutes and *k* is the apparent permeability rate constant. 

With the determined *c*_f_ the glucose transfer index (GTI)—the glucose concentration after reaching equilibrium proportional to the blank—was calculated using the equation of Espinal-Ruiz et al. [33] as follows:(4)$${\mathrm{GTI}=100\;(c}_{f}/c_{f,\mathrm{blank}})$$

#### 2.4.2. Determination of the Diffusion Coefficient

The diffusion coefficient of glucose was determined by fitting to Higuchi equation [34]:(5)$$Q/c_{0}=2\sqrt{\mathrm{Dt}}/\pi$$
where *Q* is the areal cumulative released amount [mg/cm^2^], *c*_0_ is the initial glucose concentration in the sample [mg/mL], *D* the diffusion coefficient [cm^2^/min] and *t* is the time [min].

The areal cumulative released amount of glucose was normalized using the initial glucose concentration (*Q*/*c*_0_) and plotted against the square root of time. The *D* value was then calculated from the slope of the linear regression *k*_H_ [cm/min^1/2^] using the following equation:(6)$$k_{H}=2\sqrt{D/\pi }$$

### 2.5. Rheological Investigation

#### 2.5.1. Zero-Shear Viscosity 

Rheological investigations were performed using a rotational rheometer (Physica MCR 301, Anton Paar GmbH, Graz, Austria) equipped with Rheoplus software version 3.40 (Anton Paar GmbH). Determinations were performed in at least duplicate using a concentric cylinder system (diameter: 27 mm, shear gap: 1.14 mm) (CC27-SN24807, Anton Paar GmbH) at a constant temperature of 37.0 ± 0.1 °C. The samples were pre-sheared at 2 s^−1^ for 30 s and allowed to rest for 90 s before starting the measurement.

A hysteresis curve was recorded by using a steady shear mode in logarithmic scale ranging from 0.5 to 500 s^−1^ with 10 measuring points per decade and a measurement point duration from 20 to 10 s during the forward ramp and 10 to 20 s during the backward ramp. One determination contained a forward and a backward ramp.The solutions were prepared in duplicate (Section 2.2) and analyzed twice, resulting in four separate determinations for evaluating the zero-shear viscosity as described below.

#### 2.5.2. Determination of Viscosity of CMC Solutions for Correlation to IVGR

CMC solutions for IVGR investigations (Section 2.2) were analyzed before and after performing the IVGR using a parallel plate geometry (diameter: 50 mm, shear gap: 1.00 mm) (PP50-SN23165; Anton Paar GmbH) for determining the zero shear viscosities of those solutions.

The measurements were conducted at 37.0 ± 0.1 °C and the samples were pre-sheared at 2 s^−1^ for 10 s and allowed to rest for 60 s. Viscosity flow curves were obtained in duplicate with the operating shear rate in logarithmic scale ranging from 0.5 to 500 s^−1^ with 10 measuring points per decade and a measurement point duration from 20 to 10 s.

#### 2.5.3. Curve Fitting and Determination of the Critical Concentration c*

Zero-shear viscosities were determined by fitting the data of the forward ramp to the Cross model (Equation (7)) [28] using the Rheoplus software (Anton Paar GmbH).
(7)$$\eta_{\dot{\gamma }}{=\eta }_{\infty }{+[\eta }_{0}-\eta_{\infty }]/[1+{\left(C\dot{\gamma }\right)}^{P}]$$
where *η*_0_ is the zero-shear viscosity at the lower Newtonian plateau, *η_∞_* is the viscosity at infinite high shear rate, $\dot{\gamma }$ is the shear rate and *C* is the Cross time constant or Consistency of a solution and *P* is the (Cross) rate constant. The zero shear viscosity of each solution was calculated using data from the forward ramp. Low viscous samples displaying ideal Newtonian behavior were evaluated using the Newtonian model of the Rheoplus software (Anton Paar GmbH):(8)$$\eta =\tau /\dot{\gamma }$$
where *η* and $\dot{\gamma }$ are the dynamic viscosity and the shear rate and *τ* is the shear stress. 

The viscosity is expressed as the specific viscosity (*η*_sp_), which is defined as the ratio of the viscosity of the dissolved polymer (*η*) and the solvent viscosity (*η_0_*) [17]:(9)$$\eta_{\mathrm{sp}}=(\eta -\eta_{s})/\eta_{s}$$

To obtain the critical concentration *c**, a double logarithmic plot of the zero-shear specific viscosity (*η*_sp,0_) against concentration [g/100 g] was used and data in the diluted region and the concentrated region were both fitted using a power law equation. The interception of the two fits is the critical concentration *c**.

A power Law model (Equation (10)) was used to analyze the flow curves using Rheoplus software (Anton Paar GmbH):(10)$${\dot{\gamma }=K\;\eta }^{-n}$$
where $\dot{\gamma }$ is the shear rate, *η* is the viscosity, *K* is the flow consistency index and *n* is the flow behavior index. 

#### 2.5.4. Determination of the Reynolds Number

The Reynolds numbers (*Re*)—a ratio of inertial and viscous forces—of the solutions of the stirred release experiments have been calculated:(11)$$Re=\rho {ND}^{2}/\eta_{0}$$
where *ρ* is the density of the solution, *N* is the rotational speed of the stirrer (150 rpm), *D* is the diameter of the stirrer (0.5 cm) and *η*_0_ is the zero shear viscosity of the solution.

### 2.6. Optical Density 

The optical density of CMC solutions for IVGR investigations (preparation Section 2.2) was determined using a spectrophotometer (Specord 210 plus, Analytik Jena AG, Jena, Germany). Spectral analysis of wavelengths ranging from 190 nm to 900 nm was conducted at 37 °C. The wavelength at 285 nm was chosen for the analysis, as the optical density showed the greatest deviation compared to the blank.

### 2.7. Absolute Density

The absolute densities of the CMC solutions were measured by a Gay-Lussac pycnometer (50 mL, Blaubrand^®^, Brand GmbH & Co. KG, Wertheim, Germany) at 37 ± 0.1 °C. The pycnometer was filled with sample solution, covered and left over night to ensure enclosed air bubbles to ascend. Before weighing, the filled pycnometer was equilibrated at 37 °C in a water bath for 20 min and degassed by ultrasonication at 37 °C for 3 min. The pycnometer was closed with the lid enabling excess solution to exit through a capillary and immediately weighed.

The density of the solution was calculated according to the following equation [35]:(12)$$\rho_{S}=\frac{m_{S}-m_{0}}{V}$$
where *ρ*_S_ is the density of the sample, *m*_S_ is the mass of pycnometer filled with sample, *m*_0_ is the mass of the empty pycnometer and *V* is the volume of the pycnometer. The volume of the pycnometer was determined using distilled water at 37 °C.

### 2.8. Cryo Scanning Electron Microscopy of CMC Solutions

A droplet of the highest concentrated CMC solution (3.5 × *c**) (Section 2.2) was placed in between two rivet eyelets in a sample carrier and plunged into slush nitrogen at atmospheric pressure at −205 °C. A very small droplet of the solution was used for fast freezing to avoid structural change caused by expansion. To study the microstructure, the frozen specimen was transferred to a cryo preparation unit (PolarPrep 2000, Cryo Transfer System, Quorum Technologies Ltd., Lewes, UK) and freeze-fractured at −150 °C within high vacuum (*p* = 10^−3^−10^−4^ Pa). The surface water was slightly sublimated at −80 °C for 20 to 30 min and the fractured sublimated sample was then sputter coated at −150 °C (*p* = 1.5 × 10^1^ Pa) with 3–5 nm of Pt/Pa. The coated sample was transferred into a scanning electron microscope (JSM F7200, JEOL Ltd., Tokyo, Japan) using the SE detector and the SEM mode. The images were taken with 1–5 kV accelerating voltage, probe current 5–10 and maintaining the sample below −140 °C.

### 2.9. Statistical Analysis

Results are expressed as mean ± SD. Statistical analysis was performed using SigmaPlot 12.0 for Windows (Systat Software GmbH, Erkrath, Germany). After testing for homogeneity of variance (Levene’s test) and normal distribution (Shapiro-Wilk test), one-way analysis of variances (ANOVA) was applied, and Tukey’s honestly significant difference post hoc test was used to determine the significance of differences between samples for *p* ≤ 0.01. Regressions were calculated using OriginPro 2018 for Windows (Origin Lab Corporation, Northampton, MA, USA).

## 3. Results and Discussion

### 3.1. Flow Behavior and Determination of Critical Overlap Concentration c*

#### 3.1.1. Flow Behavior of the CMC Solutions

Flow curves of the carboxymethyl cellulose (CMC) solutions in a wide range of concentrations (*c*) from 0.03 to 7.00 g/100 g were determined. Diluted solutions at low concentrations showed Newtonian behavior, exemplarily shown for CMC-H at varying concentrations from 0.03 to 0.1 g/100 g in Figure 2. At higher concentrations, a shear thinning behavior was observed, as seen for CMC-H at concentrations of 0.3 g/100 g or higher (Figure 2). 

Those curves displayed a horizontal “Newtonian plateau” at low shear rates, whereas the shear thinning behavior became obvious as shear rates increased. With increasing CMC concentrations, the onset of shear thinning shifted to lower shear rates. Shear thinning behavior appears above the onset of coil overlap, that is commonly referred to as *c** [17], which was determined between 0.1 g/100 g and 0.3 g/100 g for CMC-H (Figure 2). Furthermore, the viscosity of all solutions rose with increasing concentration of CMC as expected. The data of the forward and backward ramp overlaid for all measurements implying shear stability (data not shown). For determining zero-shear viscosities (*η*_0_), the shear thinning flow curves of the different types and concentrations of CMC were fitted to the Cross equation (Equation (7)). The pseudoplastic flow behavior of the CMC solutions agree with current data of other random coil polysaccharide flow curves [20,27,29], which also exhibit shear thinning behavior above *c**.

#### 3.1.2. Determination of *c** of the CMC Solutions

Figure 3a shows a double logarithmic plot of specific viscosity at zero shear (*η*_sp,0_) against the solution concentration of the three different types of CMC. The viscosity of the solvent (phosphate buffer, 37 °C) was 0.80 ± 0.01 mPa s, being close to the dynamic viscosity of water at 37 °C with 0.69 mPa s. The calculated critical coil overlap concentration *c** of CMC-L, CMC-M and CMC-H were 1.3 g/100 g, 0.43 g/100 g and 0.23 g/100 g linked to a *η*_0_ of 6.4, 6.3 and 6.0 mPas at the *c**, respectively. At *c* < *c** the individual coils are free to move independently and at *c* > *c** the individual coils begin to touch, overlap and interpenetrate [17], represented by a strong rise of *η*_sp,0_ with increasing concentration (Figure 3a). Most likely due to the polydispersity of CMC the increase of *η*_sp,0_ with concentration was more gradual and did not represent sharp *c** [17]. Benchabane and Bekkour [25] and Shelat et al. [21] also reported similar findings of a non-sharp *c** for CMC and arabinoxylan solutions. Wagoner et al. [27] reported a *c** of 0.67 g/100 g for a CMC with a slightly lower molecular weight (*M_W_*) of 250 kDa (DS 0.7) compared to CMC-M and with a higher *η*_0_ of 22 mPas. In contrast to those findings, Charpentier et al. [26] found a lower *c** of 1.5 g/L of a CMC with a similar molecular weight of 300 kDa (DS 0.9) compared to CMC-M in our investigation. However, the determined *c** is strongly dependent on molecular weight and substitution degree and thus direct comparisons are difficult.

Figure 3b shows a plot of *η*_sp,0_ against *M_W_* × *c.* The product of *c* and *M_W_* led to a close overlap in the double logarithmic plots of the three types of CMC. 

The slope values for the fitted regions (one for *c* ≤ *c** and the second one for *c* ≥ *c**) were 1.5 and 4.0, meaning a power law dependency of viscosity on concentration in the dilute region with *c*^1.5^ and above coil overlap with *c*^4.0^, respectively. As the second slope is approx. 2.5 times higher than the first slope, the solutions show a strong hyperentanglement and a high dependency of viscosity on concentration above *c** [17].

The product of *M_W_* × *c* describes the total degree of space occupied by the CMC in the solution, as the concentration is proportional to the number of coils present and the *M_W_* to the volume occupied by each of the CMC coils [36]. 

The observed slope values correspond quite well with the slope values of 1.4 and 3.3 for different disordered polysaccharides such as alginate, carrageenan or carboxymethyl amylose reported by Morris et al. [17]. Similar slopes were also reported for guar gum solutions with 1.5 and 4.2 [37], for beta glucan solutions of 1.3 and 4.1 [23] and for xyloglucan solutions of 1.3 and 4.0 [38]. Recently published works of Wagoner et al. [27] and Benchabane and Bekkour [25] also reported two slopes of CMC solutions with a slope of 2.7 in the concentrated region above *c** [27]. 

### 3.2. Concentration-Dependent Diffusion of Glucose

The glucose release over a time of 24 h is exemplarily shown for the CMC-L solutions with concentrations of 0.28 to 4.52 g/100 g and the solvent blank in Figure 4. The release of glucose increased with time and a reduced glucose release of the solutions with *c* ≥ *c** was visible in the decelerated rise of the curves. The blank solution showed the fastest glucose release. The equilibrium of the blank (54.1 ± 0.2 mmol/L) and samples with lower concentrations < *c** was slightly higher than the expected equilibrium of around 50 mmol/L, likely caused by a slight evaporation of water over time and hence an increased concentration of the solution.

The glucose release at equilibrium was significantly lower for the two highest concentrations of CMC-L of 2.83 g/100 g and 4.52 g/100 g with 41.4 ± 0.8 mmol/L and 25.8 ± 3.4 mmol/L, respectively, compared to the blank (54.1 ± 0.2 mmol/L). This results in a significantly lower maximum glucose transfer index (GTI) in equilibrium of 76.5 ± 1.5% and 47.8 ± 6.3% compared to the blank (100.0 ± 0.3%) (Table 2). This implies that 23.5% and 52.2% of the glucose was retarded. Consistent with the observations in Section 3.1, the rise in *c* led to an significant increase in viscosity of the solutions in a wide range from 0.8 ± 0 mPa s (blank) to 1355.3 ± 41.2 mPa s (0.8 g/100 g CMC-H) (Table 2). 

We observed a slight rise in viscosity of the same solutions before and after the release experiments (Table 2), probably caused as well by a slight evaporation of water during the 24 h. This rise in viscosity during the 24 h might have led to a stronger retardation of glucose and hence to a slightly too low measured GTI. A reduced and slowed glucose transfer is most likely caused by the rise in viscosity with increasing *c* of the different types of CMC, which has already been described previously [10]. For example, a concentration increase from the blank to 4.52 g/100 g CMC-L in solution led to a significant rise in viscosity from 0.8 ± 0.0 to 519 ± 9.1 mPa s and a significant decrease in the maximum GTI from 100.0 ± 0.3 to 47.8 ± 6.3%, respectively (Table 2). This rise in viscosity also came along with a significant reduction in the diffusion coefficient (*D*) of glucose from 1.42 ± 0.04 *×* 10^−8^ to 0.16 ± 0.02 *×* 10^−8^ m^2^s^−1^, respectively (Table 2). Similar observations with retarded glucose diffusion and lowered amount of glucose transfer due to increased viscosity were reported by Espinal-Ruiz et al. [33] for pectin, by Ou et al. [4] for different soluble and insoluble fibers including CMC and by Srichamroen and Chavasit [5] for malva nut gum.

The reduced GTI can be caused by an entrapment of the glucose molecules by a highly concentrated and viscous fiber network and as well by a chemical bonding of glucose molecules to the fibers [4,10]. However, as glucose is electrically neutral at pH 7.0, electrostatic interactions with the different dietary fibers were considered unlikely (Espinal-Ruiz et al., 2016), as chemical bonding—except weak hydrogen bonds—of glucose to fiber molecules is more likely to occur in insoluble fibers [4,39,40].

### 3.3. Correlations between Diffusion Coefficients, Concentrations and Molecular Weight

A linear dependency of *D* with both the concentration and *M_W_* of the different types of CMC was observed (Figure 5). The solvent blank exhibited along with a concentration of 0.05 g/100g of CMC-H the highest *D* of 1.42 ± 0.04 *×* 10^−8^ m^2^ s^−1^ (Table 2). No significant differences (*p* ≤ 0.01) were observed between the *D* of the solutions below *c** and the *D* in the blank solution (Table 2). In solutions with concentrations ≥*c** the *D* was significantly lower (*p* ≤ 0.01) for all types of CMC compared to the *D* in the blank solution, which indicates that for impairing glucose release the knowledge of *c** is important and that only concentrations of CMC higher than *c** significantly influenced in vitro glucose release. 

Moreover, there was a trend that the diffusion coefficients of CMC-H were highest, followed by CMC-M and CMC-L, respectively, when comparing the same proportional *c* × *M_W_* (Figure 5). Contrary the viscosities were even higher for CMC-H and CMC-M compared to CMC-L at the given *c* × *M_W_* (Table 2). For example, at the highest *c* × *M_W_* of each sample, CMC-M and CMC-H showed higher *D* (0.56 ± 0.02 and 0.53 ± 0.01 *×* 10^−8^ m^2^ s^−1^, respectively) compared to CMC-L with the lowest *D* of 0.16 ± 0.02 *×* 10^−8^ m^2^ s^−1^. The viscosity at the given *c* × *M_W_* of the solution (Table 2) was 1.6 and 2.6 times higher (Table 2) for CMC-M and CMC-H compared to CMC-L, respectively. 

These results indicate that soluble fibers with a low *M_W_* in higher *c* are more effective in retarding glucose diffusion than high *M_W_* soluble fibers in lower *c*. Similarly, Shelat et al. [21] observed as well at a given concentration of two types of arabinoxylans the solution with the arabinoxylan forming a higher viscosity showed a higher probe *D* compared to the solution with the arabinoxylan forming a lower viscosity. Shelat et al. [20,21] and Liu et al. [40] observed also a decrease of *D* with an increase of polysaccharide concentration ranging from 0.01–1.00% (*w*/*v*) (arabinoxylan), from 0.5–1% (*w*/*v*) (β-glucan) and from 0.01–2% (*w*/*w*) (nano-fibrillated cellulose) respectively. Unlike our results, the authors observed an initial sharp decrease of *D* with an increase of *c* resulting in an exponential shaped curve. Han et al. [41] observed an initial increase in diffusion of aspartame in very low concentrated CMC solutions and consistent with our results a decrease in *D* with concentrations higher than *c** due to the formation of an entangled network. The positive influence on glucose diffusion of solutions with concentrations of CMC above the coil overlap (>*c**.) further supports the findings of Rieder et al. [23] of a significant in vivo effect of *c** on the post-prandial blood glucose rise as shown by in-depth meta-analysis on glucose responses to β-glucan samples. 

To reach concentrations higher than the *c** in the small intestine, it has to be taken into account that the fiber containing meal might be strongly diluted depending on the composition of the more complex and heterogeneous meal. According to Minekus et al. [30] the dilution factor can be up to eight. Therefore, the concentration of fiber in the food has to be adopted in order to remain the impact of diffusion in the small intestine [3].

### 3.4. Correlations between Diffusion Coefficients and Viscosity—Deviation from Stokes-Einstein Equation

The *D**η/D*_0_*η*_s_ ratio demonstrates the quantitative deviation from the Stokes-Einstein (SE) behavior (Equation (1)), whereby *D* and *D*_0_ are the diffusivities of glucose in the sample and blank solution, respectively, and *η* and *η*_s_ are the viscosities of the sample and blank solution, respectively. 

Figure 6 shows a positive deviation from Stokes-Einstein behavior with increasing *c* × *M_W_* up to over 615 ± 166 for CMC-H and 394 ± 71 and 71 ± 1 for CMC-M and CMC-L, respectively. The positive deviation of *D**η/D*_0_*η*_s_ > 1 implies that the decrease in diffusivity is less strong compared to the increase in viscosity of the solution [19]. 

The sharp increase at the *c** with deviations of 15 ± 4, 12 ± 2 and 7 ± 1 for CMC-H, CMC-M and CMC-L, respectively, is visible and refers to the strong increase in viscosity due to the entanglement of the CMC polymers. Shelat et al. [20,21] measured the diffusivity of a dextran probe in arabinoxylan and β-glucan solutions at different concentrations ranging from 0.2–1% and 0.5–1%, respectively, and determined as well a positive deviation from Stokes-Einstein relation up to app. 50. For comparability, it has to be taken into account that the *M_W_* of the probe of the authors is almost 400 times higher than the *M_W_* of glucose used in our work.

For small spheres as glucose and large chains as present in our study, the positive deviations can be explained by an easier movement of the glucose through the solution even with the CMC chains still tensed in conformation and forming a high viscous network [19]. 

Another possible reason is an inhomogeneous formation of an aggregated network by forming regions with high and low concentrations of CMC, faster for glucose to move through [21].

In addition, the positive deviations could also originate from a decrease in the hydrodynamic radius of glucose with increasing concentration of CMCs. This is due to interactions between glucose and the CMC molecules [21]. However, as the total degree of space occupied by the CMC is proportional to *M_W_* × *c* (Section 3.1) and consequently proportional to the amount of water bound to the CMC, the decrease in the hydrodynamic radius of the glucose molecules in the solutions with similar *M_W_* × *c* should be effectively equal. Hence, the influence of a decreased hydrodynamic radius of glucose on the diffusivity in solutions with similar *M_W_* × *c* should be negligible.

Figure 7 shows cryo scanning images of the three different types of CMC in solutions at their highest concentrations (4.5 g/100 g for CMC-L, 1.5 g/100 g for CMC-M, 0.8 g/100 g for CMC-H). It is visible that the shorter chains of CMC-L in a higher concentration formed a solution in a higher density with less free spaces compared to the longer chains of CMC-M and CMC-H in a lower concentration forming a solution with more free spaces. The higher concentration of the short chain molecules hamper the glucose molecules in their diffusion through the solution in a greater extent than a lower concentration of the long chain molecule at a similar given viscosity.

Fujita [42] implemented the exponential dependence of the diffusivity of drugs on the free volume in polymers, where a more difficult diffusion in the CMC-L solution is most likely. This corroborates our finding of a lower *D* of the CMC-L solutions compared to CMC-H and CMC-M solutions and further the observation of the more positive deviations from SE behavior for the CMC-H and CMC-M solutions compared to CMC-L solutions, leading to a high diffusivity despite high viscosity.

The optical densities of the three solutions at a wavelength of 285 nm also confirmed the observation of the SEM images with a rise in density of 0.46 ± 0.02, 0.51 ± 0.01 and 1.4 ± 0.04 for the highest concentrated solutions of CMC-H (0.8 g/100 g), CMC-M (1.5 g/100 g) and CMC-L (4.5 g/100 g), respectively (Table 2). Liu et al. [43] detected CMC absorption in aqueous solutions at a similar wavelength, probably involving the stimulation of carboxylic acids [44].

Nsor-Atindana et al. [15] also observed differences in retarding glucose and diffusion depending on the size of nanocrystalline cellulose, whereby the smallest particles were the most effective in attenuating glucose diffusion. 

The diffusivity of glucose and the viscosity—as the macroscopic behavior of the solution—are not necessarily correlated [21] and other microscopic properties are able to affect the diffusivity as the *M_W_* of the fiber and/or the density of the entangled network in the solution. Therefore, diffusivity and viscosity of a solution should be determined independently.

### 3.5. Impact of Convection

In addition to the previously described diffusion experiments, we also investigated the impact of viscosity on glucose release under convection conditions by applying constant stirring (150 rpm) of the donor cell. The release experiments were conducted with the highest concentrated CMC solutions (4.5 g/100 g for CMC-L, 1.5 g/100 g for CMC-M, 0.8 g/100 g for CMC-H) and the results are given in Figure 8. 

Stirring increased glucose release and the maximum GTI for all CMC solutions except the blank, indicating a limitation of glucose release by the membrane. The Reynolds numbers (*Re*) of the solutions (Table 3) were calculated according to Equation (11) using *ρ* and *η*_0_ of the solutions listed in Table 3.

The highest acceleration of the released glucose by convection has occurred in the CMC-L solution (maximum GTI from 47.8 ± 6.3 to 111.2 ± 3.2%) with the highest *Re* of the three types of CMC in solutions, referring to the highest mixing of the solution (Table 3). The CMC-H solution had the strongest minimization of the *Re* by a factor of approx. 1500 compared to the blank leading to a more than 2.5 times lower *Re* compared to the CMC-L solution. Nevertheless, the glucose release was comparable with the CMC-L solution and not lower as expected. One reason could be the lowest *n* of 0.66 ± 0.01 of CMC-H of the three solutions, indicating a strong dependence of viscosity on the shear rate, leading to an around 8-fold drop in viscosity by an increase of the shear rate from 0 to 158 s^−1^. This decrease in viscosity could result in a faster release of glucose. 

The high *Re* of CMC-L solution could be caused by a better mobility of the short CMC-L chains. This led to a high glucose acceleration with stirring because glucose can move without waiting for the CMC-L chains to relax their conformation. Since convection in the solution increased the maximum GTI of all types of CMC solutions, we assume that a lowered maximum GTI is more likely caused by entrapped glucose in the viscous matrix than chemical bonding.

The effects of convection on glucose release were also reported by Liu et al. [40]. The retardation of glucose by a 2% (*w*/*w*) nano-fibrillated cellulose solution dropped from 50% to 27.7% due to shaking. Dhital et al. [45] observed that mixing at different speeds (0, 200 and 750 rpm) significantly accelerated the mass transfer of glucose out of barley beta-glucan solutions. The results show that the ability of CMC to form high viscous solutions could inhibit turbulence, induced by peristalsis, by reducing the Reynolds numbers in the gut content and provide laminar flow, leading to a reduced rate of glucose absorption [9].

## 4. Conclusions

The main objective of this work was to investigate the impact of the parameters molecular weight, concentration—particularly the critical concentration (*c**)—and viscosity of CMC in solutions on glucose release using an in vitro side-by-side system. Our hypothesis of a positive influence on glucose diffusion of solutions with concentrations of CMC above the coil overlap (≥*c**.) was confirmed and the results underline physiological outcomes of in vivo studies. Our results indicate that chyme in a higher concentration of lower molecular weight (*M_W_*) fibers leads to a stronger decrease in diffusion of glucose from the gut lumen to the enterocytes then a lower concentration of higher *M_W_* fibers, at a similar given viscosity. The outcome of our study provides an excellent basis to elucidate the occurrence of the strong deviation from Stokes-Einstein behavior in fiber solutions. The subject of future research should comprise further studies on additional influencing factors of dietary fibers on glucose diffusion in order to improve the overall understanding of the health effects of dietary fibers.

## Figures and Tables

**Figure 1 nutrients-13-01398-f001:**
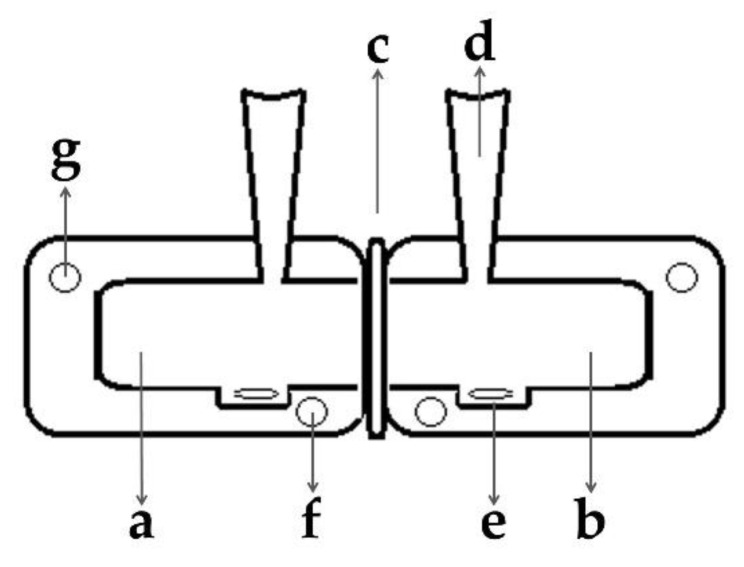
Schematic of the side-by-side diffusion system. Heatable double wall donor cell with a nominal volume of 4 mL (**a**), heatable double wall receptor cell with a nominal volume of 4 mL (**b**), position of the semipermeable cellulose membrane (**c**), sampling port (**d**), magnetic stir bar (**e**), water inlet (**f**) and water outlet (**g**).

**Figure 2 nutrients-13-01398-f002:**
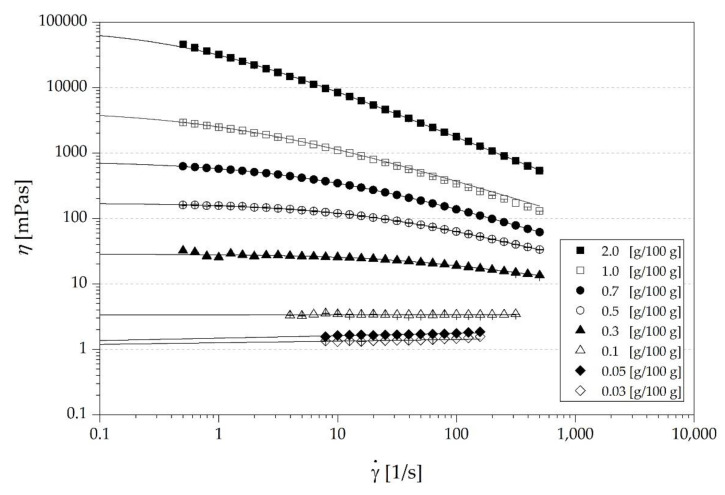
Flow behavior exemplarily shown for the high *M_W_* carboxymethyl cellulose solutions (CMC-H) at different concentrations [g/100 g]. Shear rate dependent viscosity appeared for solutions in concentrations above *c** (0.3 g/100 g–2.0 g/100 g). Points correspond to experimental data and lines to fitted data according to Cross equation (Equation (7)) for concentrations of 0.3–2.0 g/100 g and Newtonian equation (Equation (8)) for concentrations of 0.03–0.1 g/100 g (R^2^ ≥ 0.99).

**Figure 3 nutrients-13-01398-f003:**
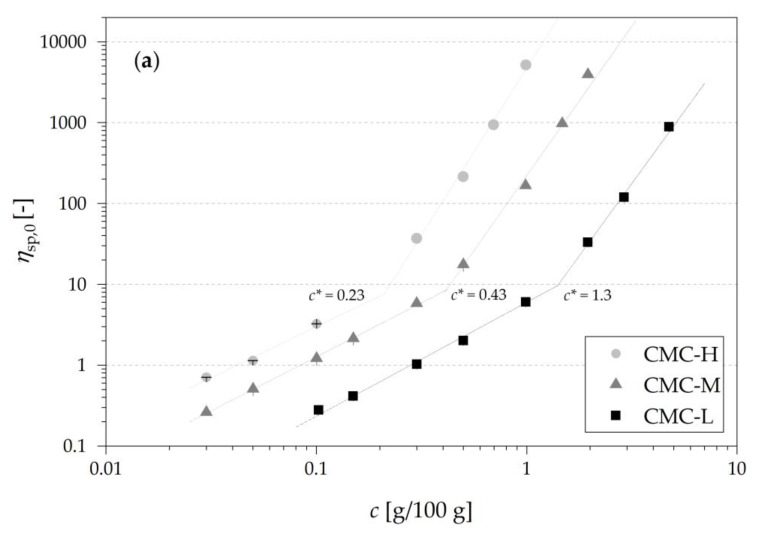
Double logarithmic plot of zero shear specific viscosity against concentration *c* (**a**) and against the product of *c* and *M_W_* (**b**) of three types of CMCs with different *M_W_*: CMC-L (100 kDa), CMC-M (395 kDa), CMC-H (725 kDa). Data in the diluted regions and the concentrated regions were both fitted using a power law equation (R^2^ ≥ 0.95).

**Figure 4 nutrients-13-01398-f004:**
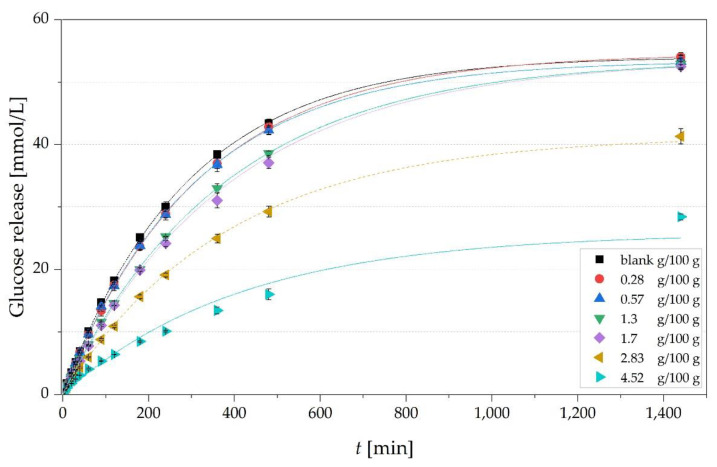
Time-dependent glucose release from solutions with different concentrations of CMC-L (0.28–4.52 g/100 g; blank = 0 g/100 g). 1.3 g/100 g is the calculated *c** concentration of CMC-L. Points correspond to experimental data and lines to fitted data according to equation 3 (R^2^ ≥ 0.9998).

**Figure 5 nutrients-13-01398-f005:**
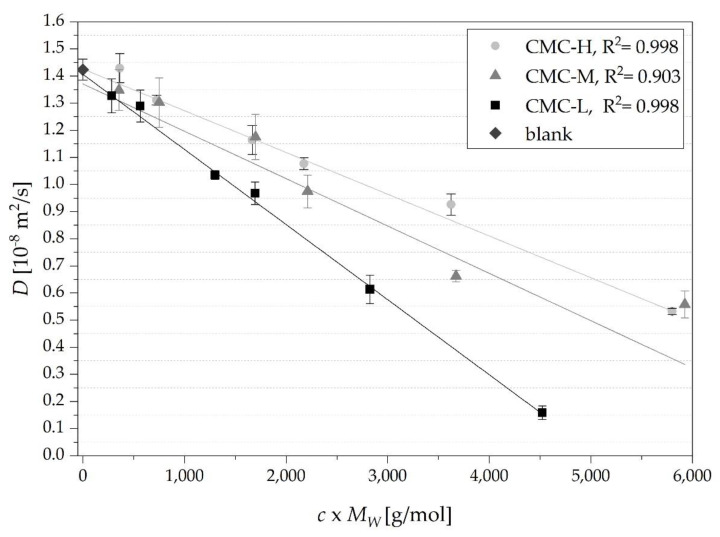
Correlation of diffusion coefficient of glucose in CMC solutions and increasing concentrations × *M_W_* of the three different CMC types: CMC-L (100 kDa), CMC-M (395 kDa) and CMC-H (725 kDa). Points correspond to experimental data and lines to fitted data using linear regression (R^2^ ≥ 0.903).

**Figure 6 nutrients-13-01398-f006:**
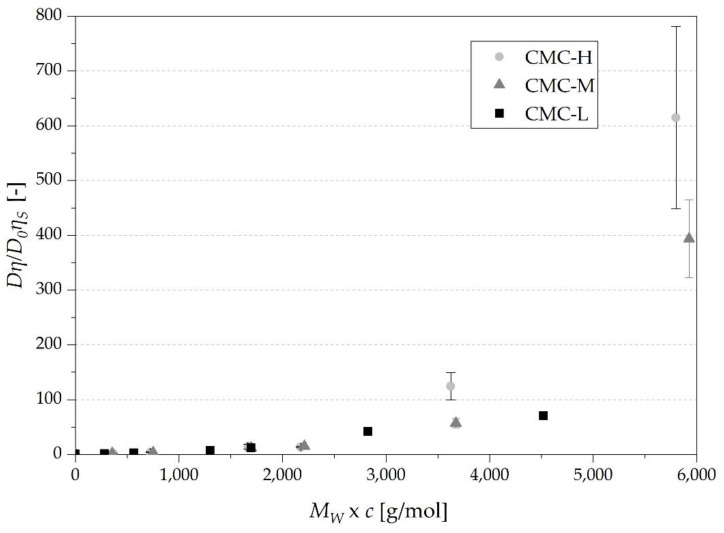
Deviation from Stokes-Einstein equation (Equation (1)) of the three different types of CMC. The calculated *M_W_* × *c** for CMC-L, CMC-M and CMC-H were 130,000, 169,850 and 166,750 g/mol, respectively.

**Figure 7 nutrients-13-01398-f007:**
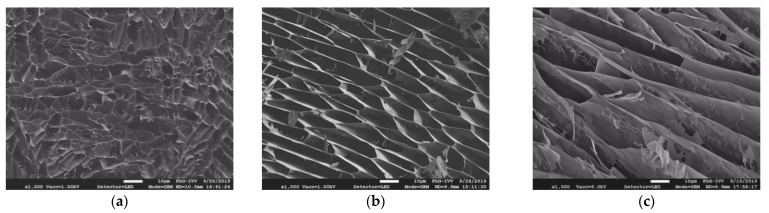
Cryo scanning electron microscope images of the three CMC solutions in their highest concentration of 3.5 × *c** at the same scale and a magnification of 1000: 4.5 g/100 g CMC-L (**a**), 1.5 g/100 g CMC-M (**b**) and 0.8 g/100 g CMC-H (**c**).

**Figure 8 nutrients-13-01398-f008:**
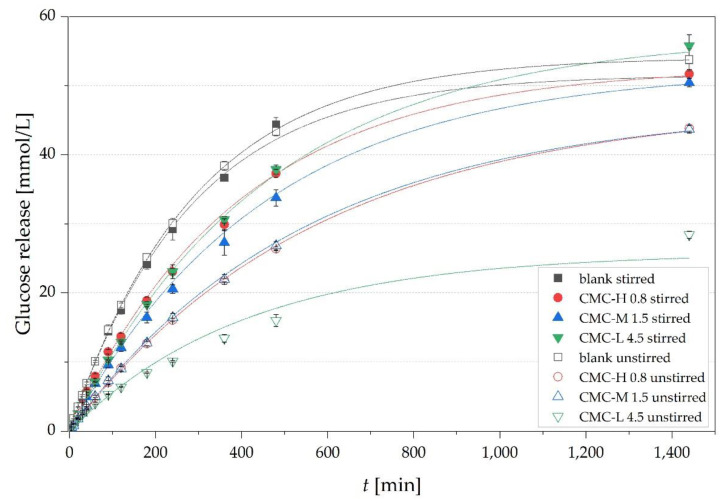
Time-dependent glucose release from CMC-H, CMC-M and CMC-L at their highest concentration of 0.8 g/100 g, 1.5 g/100 g and 4.5 g/100 g, respectively, and the blank with and without stirring (150 rpm). Points correspond to experimental data and lines to fitted data according to equation 3 (R^2^ ≥ 0.9998).

**Table 1 nutrients-13-01398-t001:** Fiber concentrations for the in vitro glucose release test.

Calculation	Concentration [g/100 g]		
	CMC-L	CMC-M	CMC-H
*c*_0_ = 0	0	0	0
*c*_1_ = 0.22 × *c**	0.28	0.09	0.05
*c*_2_ = 0.43 × *c**	0.57	0.19	0.10
*c**	1.30	0.43	0.23
*c*_4_ = 1.30 × *c**	1.70	0.56	0.30
*c*_5_ = 2.17 × *c**	2.83	0.93	0.50
*c*_6_ = 3.48 × *c**	4.52	1.50	0.80

CMC-L: low-*M_W_* CMC; CMC-M: medium-*M_W_* CMC; CMC-H: high-*M_W_* CMC. *c** presents the critical concentration.

**Table 2 nutrients-13-01398-t002:** Zero-shear viscosity before and after 24 h release experiment, diffusion coefficient and optical density at different concentrations around the *c** of CMC-L, CMC-M, CMC-H.

Concentration	*η*_0_ [mPa s]	*η*_0_ after 24 h [mPa s]	Diffusion Coefficient (*D*) [10^−8^ m^2^s^−1^]	Maximum GTI [%]	Optical Density at 285 nm
0 × *c** (blank)	0.8 ± 0.0 ^a^	0.8 ± 0.0 ^a^	1.42 ± 0.04 ^a^	100.0 ± 0.3 ^a^	0.17 ± 0.0 ^a^
CMC-L					
0.2 × *c** (0.28 g/100 g)	1.5 ± 0.0 ^a^	1.6 ± 0.0 ^a^	1.33 ± 0.06 ^a,b^	100.9 ± 0.5 ^a^	0.23 ± 0.0 ^a^
0.4 × *c** (0.57 g/100 g)	2.6 ± 0.1 ^a^	2.7 ± 0.0 ^a^	1.29 ± 0.06 ^a,b^	98.7 ± 0.6 ^a^	0.28 ± 0.03 ^a^
*c** (1.30 g/100 g)	8.3 ± 0.1 ^a^	8.8 ± 0.2 ^a^	1.03 ± 0.02 ^c^	99.1 ± 1.0 ^a^	0.48 ± 0.02 ^b^
1.3 × *c** (1.70 g/100 g)	14.5 ± 0.2 ^b^	15.3 ± 1.1 ^a^	0.97 ± 0.04 ^c^	99.3 ± 0.5 ^a^	0.55 ± 0.07 ^b^
2.2 × *c** (2.83 g/100 g)	78.7 ± 2.0 ^c^	66.9 ± 5.4 ^b^	0.61 ± 0.05 ^d^	76.5 ± 1.5 ^b^	0.88 ± 0.02 ^c^
3.5 × *c** (4.52 g/100 g)	519 ± 9.1 ^d^	559.5 ± 16.1 ^c^	0.16 ± 0.02 ^e^	47.8 ± 6.3 ^c^	1.4 ± 0.04 ^d^
CMC-M					
0.2 × *c** (0.09 g/100 g)	1.7 ± 0 ^a^	1.8 ± 0.1 ^a^	1.35 ± 0.08 ^a,b^	101.5 ± 0.7 ^a^	0.18 ± 0.0 ^a^
0.4 × *c** (0.19 g/100 g)	3.4 ± 0.1 ^a^	2.9 ± 0 ^a^	1.3 ± 0.09 ^a,b^	98.3 ± 0.9 ^a^	0.2 ± 001 ^a^
*c** (0.43 g/100 g)	11.3 ± 0.4 ^a,b^	11 ± 0.2 ^a^	1.18 ± 0.08 ^b,c^	97.0 ± 1.8 ^a^	0.25 ± 0.0 ^b^
1.3 × *c** (0.56 g/100 g)	17.9 ± 0.1 ^b^	19.5 ± 0.4 ^a^	0.97 ± 0.06 ^c^	89.6 ± 1.0 ^b^	0.28 ± 0.01 ^b^
2.2 × *c** (0.93 g/100 g)	99.8 ± 3.1 ^c^	99.1 ± 6.3 ^b^	066 ± 0.02 ^d^	88.0 ± 1.2 ^b^	0.35 ± 0.03 ^c^
3.5 × *c** (1.50 g/100 g)	816.2 ± 12.6 ^d^	1010.2 ± 18.0 ^c^	0.56 ± 0.02 ^d^	86.5 ± 1.3 ^b^	0.51 ± 0.01 ^d^
CMC-H					
0.2 × *c** (0.05 g/100 g)	1.9 ± 0.1 ^a^	1.7 ± 0.0 ^a^	1.43 ± 0.05 ^a^	102.7 ± 3.3 ^a^	0.18 ± 0.0 ^a^
0.4 × *c** (0.10 g/100 g)	3.5 ± 0.5 ^a^	3.7 ± 0.2 ^a^	1.31 ± 0.02 ^a^	102.5 ± 1.0 ^a^	0.2 ± 0.0 ^a^
*c** (0.23 g/100 g)	14.9 ± 1.2 ^a^	15.2 ± 1 ^a^	1.16 ± 0.05 ^b^	98.6 ± 1.1 ^a^	0.24 ± 0.0 ^b^
1.3 × *c** (0.30 g/100 g)	15 ± 0.7 ^a^	15.1 ± 0.3 ^a^	1.08 ± 0.02 ^b^	97.8 ± 1.8 ^a^	0.26 ± 0.0 ^b^
2.2 × *c** (0.50 g/100 g)	159.2 ± 4.1 ^b^	165.1 ± 3.1 ^b^	0.93 ± 0.04 ^c^	94.7 ± 1.2 ^b^	0.32 ± 0.01 ^c^
3.5 × *c** (0.80 g/100 g)	1355.3 ± 41.2 ^c^	1744.5 ± 30 ^c^	0.53 ± 0.01 ^d^	87.3 ± 2.0 ^c^	0.46 ± 0.02 ^d^

Different superscript letter (a–e) within a column indicate significant difference between means (*p* < 0.01).

**Table 3 nutrients-13-01398-t003:** Power Law parameters (obtained by Power Law model Equation (10); R^2^ ≥ 0.941), viscosities with shearing, Reynolds numbers (*Re*) and densities of the three different CMC solutions at the highest concentration (4.5 g/100 g for CMC-L, 1.5 g/100 g for CMC-M, 0.8 g/100 g for CMC-H); the calculated *Re* of the blank was 76.53 ± 1.69; viscosities at zero shear see Table 2.

Sample	Maximum GTI [%]	*K* [Pa s]	*n* [-]	$\eta (\dot{\gamma }$ = 158 s^−1^) [mPa s]	*Re* [-]	Density [kg/m^3^]
CMC-L 4.52 g/100 g	111.2 ± 3.2 ^a^	0.5 ± 0.01 ^a^	0.91 ± 0 ^a^	320 ± 6.3 ^a^	0.13 ± 0 ^a^	1027 ± 0.8 ^a^
CMC-M 1.5 g/100 g	101.6 ± 1.4 ^a^	0.76± 0.02 ^b^	0.77 ± 0 ^b^	221.5 ± 4 ^b^	0.08 ± 0.02 ^a,b^	1011.5 ± 2.1 ^b^
CMC-H 0.8 g/100 g	102.3 ± 1.6 ^a^	1.09 ± 0.04 ^c^	0.66 ± 0.01 ^c^	175 ± 4.3 ^c^	0.06 ± 0.02 ^b^	1007.9 ± 0.7 ^b^

Different superscript letter (a–c) within a column indicate significant difference between means (*p* < 0.01).

## Data Availability

The data presented in this study are available on request from the corresponding author within the terms of the contract that funded the study. Funding details are listed above.

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
