# Peer review of "Effect of Physicochemical Properties of Carboxymethyl Cellulose on Diffusion of Glucose"

_nutrients, 2021, doi:10.3390/nu13051398_

Round 1

Reviewer 1 Report

Dear Editor, 

I had the pleasure to read the manuscript “Effect of physiological properties of CMC on diffusion if glucose” which is considered for publication in the journal Nutrients.   This is an excellent piece of work which advances our mechanistic understanding of how soluble dietary fibres might reduce post-prandial glycaemic response in vivo.  I recommend the work is published without delay.  However, before the work is published the authors should consider the following comments.

  1. Line 83.  What is meant by ‘natural fluctuations’?  This description is rather ambiguous.  Can you be more specific?
  2. Line 84. ‘achieve particular effects”.  Can you be perhaps more specific?
  3. Line 99 and other places. The term ‘average molecular weight’ is used. This is in fact incorrect unless it is clear which kind of average-molecular weight one is referring to?  I’m assuming it is weight-average you a referring to?  Please correct throughout the manuscript. 
  4. Do you have any information for polydispersity of your CMC samples?
  5. Line 161. Correct ‘puffer’ (German I assume?) to ‘buffer’.
  6. Line 215. It is stated that there were four separate determinations for calculation of zero-shear viscosity for duplicate samples.  Yet, in line 227 is says only the forward ramp was fitted with the Cross equation.  S0 I take that then to mean just two separate determinations? Please correct so it is clear.
  7. Equation 10.  How did you manage to fit y=ab+x to obtain a straight line in a double-logarithmic plot?  Shouldn’t it be a power law equation where for Newtonian type samples the exponent is close to 1?
  8. Section 2.9 Statistical analysis. You state that data was analysed in OriginPro and statistical analysis performed in SigmaPlot 12.   Isn’t statistical analysis also part of data analysis?  Please make what you have done clear.   For ANOVA did you check your data set was normally distributed and with roughly equal variances?   Any transformations necessary?
  9. Should it rather be at ‘concentrations of 0.3g/100g or higher’ and not ‘ higher than 0.3 g/100g)?
  10. It is stated that a Newtonian plateau is observed in Figure 2 for otherwise shear thinning samples at low shear rates.   I disagree.   I don’t see any clear Newtonian plateaux for any samples.   This is probably because of a lack of sensitivity to measure ultralow torques by the rheometer with the CC-27 measuring geometry that was used.  In line 312 it is stated that the Cross equation has a fit of R2 = 0.99.  Please show the Cross fit on the data in Figure 2.   In this way you can extrapolate the fit to lower shear rates and observe the plateaux and in fact more importantly convince the reader that application of the cross equation can accurately calculate zero shear viscosity correctly from your dataset even when there is a lack of a clear Newtonian plateau in the fitted dataset.   
  11. Line 345. It is stated a linear regression (y=ab+x) is applied to the double logarithmic plot in Figure 3.  This should rather be a power law equation where the ‘linear slope’ is actually the exponent? 
  12. Line 319: Please delete the sentence, “To clear the direct contribution…..” It is already defined what specific viscosity means earlier in the manuscript.
  13. What was the calculated rate constant and released glucose concentration at infinity for each sample for the fits of equation 3 shown in Figure 4 and 8? I suggest this might be useful information to include in the paper. 
  14. Table 3. Should at least be all on the same page!  I guess this is just for the peer review version?   It is very difficult to see the trends in a large sea of numbers.  I suggest this data could be more clearly presented in the form of a bar chart with error bars.  

Reviewer 2 Report

This manuscript details a study investigating the importance of polysaccharide molecular weight and concentration on rheological properties and glucose diffusion in vitro. The results of this elegant study demonstrate significant associations between the examined parameters and viscosity and solute permeability. There are several minor issues that need to be addressed, outlined below.

Details:

Line 35: remove “beneficial” and change “soluble dietary fibers (SDF) including the reduction of post-prandial…

Line 47: “…and resulting in reduced post-prandial plasma glucose levels in 47 the case of glucose absorption.” Provide a reference.

Line 75: here, and throughout the manuscript, the authors use the term “glucose release”. Without further experimental evidence demonstrating binding of glucose to CMC, perhaps “glucose permeability” or “glucose transfer” is a more appropriate term…

Line 99: I assume these are the average molecular weights? How broad is the distribution?

Line 101: possible change “substitution degree” to “degree of methylation”?

Line 110: please change NaN3 to Sodium Azide.

Figure 2: I assume the crosses in the open symbols are error bars? Perhaps they are not necessary in the case…

Line 351-553: this sentence should be moved the beginning of this paragraph (line 338).

Line 374: (R2 ≥ 0.9998) to (R2 ≥ 0.9998).

Line 375: Is this really equilibrium? Are you suggesting an interaction (binding) between the glucose and the CMC?

Line 404-407: Could the authors suggest what caused the decreased glucose transfer at "equilibrium"? or is equilibrium well beyond the time scale of the experiment?

Figure 5: The linear regression for CMC-M doesn't appear to go through the blank. r2 may be higher if it does....

Figure 6: Would plotting Dη/D0ηs [-] on a log scale improve the readability of the lower values?

Line 456: Sentence is not clear…“The positive deviation of Dη/D0ηs > 1 implies that the decrease in diffusivity is reduced, compared to the increase in viscosity of the solution [19]”.  Is this correct?

Line 476:  change “by” to “for”.

Line 478:  change “origin” to “originate”.

Line 492:  change “stronger” to “greater”.

Line 512:  change “as a” to “and”.

Of importance to the Nutrients readership would be the implications of this work in real-world systems that should be covered in more detail, in particular:

  • The impact of the heterogeneity of chyme, such as the impact of particulates.
  • Dilution effects throughout digestion, i.e. at what concentrations would fibres (in this case CMC) need to be consumed to reach c* in the small intestine?
  • Apart from free glucose in the diet, the major component of postprandial plasma glucose rises are the result of starch digestion. Pancreatic a-amlyase catalyses the breakdown to maltose, maltotriose and low Mw dextrins (with further breakdown to glucose occurring ate the epithelial surface). Could the authors provide more discussion of the implications of solute Mw?
